# WHICH TRANSFORMER COMPONENTS ARE THE MOST SENSITIVE TO DISTRIBUTION SHIFTS?

## ABSTRACT

Transformers have become the default backbone of large foundation models, achieving state-of-the-art results in natural language processing, computer vision, and time series analysis. These general-purpose models are typically finetuned by practitioners on specific tasks and domains. While most methods focus on reducing the computational cost of adapting ever-larger models, a complementary stance is to better understand how the transformer architecture responds to distribution shifts – an avenue that can improve efficiency and performance. In this work, we propose an approach to study the sensitivity of transformer components to distribution shifts. By viewing sequences of tokens as discrete measures, we show that transformer encoders can be decomposed into measure-to-measure maps and define the sensitivity to distribution shifts based on an averaged notion of Lipschitz continuity, commonly associated with robustness in the literature. Through a comprehensive empirical investigation on large vision transformers (ViT) across 30 corrupted versions of ImageNet, we demonstrate that attention and feedforward normalization layers are consistently the most sensitive to input perturbations. While we do not observe that increased sensitivity steadily leads to better finetuning performance across all blocks, it is remarkably the case for the feedforward normalization layer that is both highly sensitive and matches or surpasses full finetuning while reducing the number of trainable parameters by a factor of 5000. Overall, our findings provide new insights into how transformer components behave under distribution shifts, showcasing that a better understanding of the transformer architecture can inform the design of more efficient adaptation methods.

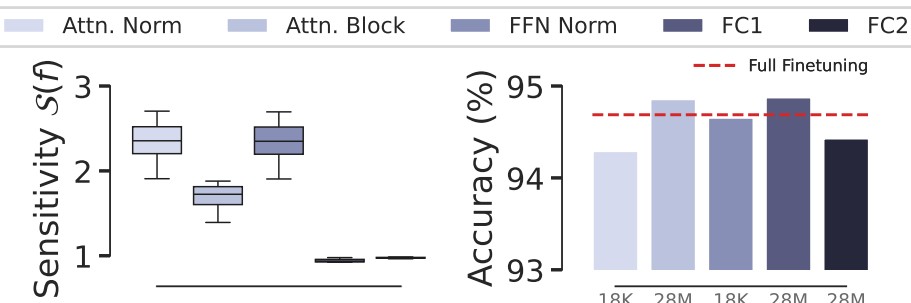

Figure 1: **Transformer sensitivity to distribution shifts.** *Left*: Sensitivity to distribution shifts $\mathcal{S}(f)$ based on the averaged notion of Lipschitz continuity (Definition 3.1) for each of the trainable components $f$ of a vision transformer pretrained on ImageNet-21k (ViT-Base) over 30 corrupted versions of ImageNet. *Right*: Performance after finetuning each trainable component in isolation for the same model averaged over 6 common image classification benchmarks with the corresponding number of trainable parameters. While the Lipschitz continuity criterion does not explain potential finetuning gains for all the components, it does so for the normalization layers that are both the most sensitive and highly competitive when seen through the lens of performance gain per parameter point of view.

# 1 INTRODUCTION

Transformers (Vaswani et al., 2017) have become the default backbone of the strongest foundation models, achieving state-of-the-art results in a wide range of applications such as natural language processing tasks (Radford et al., 2019; Brown et al., 2020), computer vision (Caron et al., 2021; Dosovitskiy et al., 2021) through time series forecasting (Ansari et al., 2024; Woo et al., 2024; Ilbert et al., 2024) or complex reasoning (Comanici et al., 2025; OpenAI et al., 2024). Those large foundation models pretrained on huge amounts of data are typically used for more specific tasks, either in a zero-shot fashion or by adapting their weights. This assumes that the knowledge integrated in the weights during pretraining can serve as a useful prior on new data (note that this assumption need not hold in the context of agentic large language models (Lewis et al., 2020; Schick et al., 2023; Houliston et al., 2025)). In practice, the discrepancy between the training and the downstream data can harm the final performance. While a lot of effort was put into reducing the cost of using bigger and bigger models for downstream tasks (Hu et al., 2022; Han et al., 2024; Ben Zaken et al., 2022; HuggingFace, 2025; VLLM, 2025), better understanding how pretrained transformers reacts to distribution shifts could improve the efficiency and quality of adaptation by modifying only the sensitive to the shifts parts of the model. This motivates us to ask:

*Which transformer components are the most sensitive to distribution shifts?*

Since foundation models can be applied to a wide range of downstream tasks and data modalities, it is important to study their sensitivity to distribution shifts at the level of the hidden representations. This, in turn, requires defining a notion of distance between sequences of tokens in order to quantify how outputs of a given component vary under input shifts. However, establishing such a distance is challenging: standard Euclidean tools are unreliable (Aggarwal et al., 2001) and prior similarity metrics between hidden representations—such as CKA (Kornblith et al., 2019)—have been shown to be sensitive to outliers and can yield counterintuitive results (Davari et al., 2023). By viewing sequences of tokens as discrete measures, we show that transformers can be decomposed into measure-to-measure maps. This allows us to use a natural distance between probability measures, the Wasserstein distance, which captures the geometry of the underlying data. We then define the sensitivity to distribution shift as an average notion of Lipschitz continuity (Definition 3.1), which is commonly associated with robustness in the literature (Rosca et al., 2020; Goodfellow et al., 2015; Miyato et al., 2016; Tsuzuku et al., 2018; Gao et al., 2023; Rosca et al., 2020). Having defined a notion of sensitivity to distribution shifts, a natural question it can translate to potential gains when finetuning pretrained transformers on out-of-distribution data.

**Main contributions.**   We summarize our contributions and findings below.

1. **Method.** We propose an approach to study the sensitivity of transformer components to distribution shifts based on an averaged notion of Lipschitz continuity, commonly associated with robustness in the literature.

2. **Analysis.** We provide a comprehensive empirical investigation of the sensitivities of the different blocks of large vision transformers (ViT) across 30 corrupted versions of ImageNet. We identify a remarkably consistent ranking across the sensitivities across transformer blocks: layer norms are the most sensitive ones, followed by attention, followed by feedforward layers. To the best of our knowledge, such a complete study has never been done in the literature before.

3. **Findings.** Despite the observed consistent behavior, increased sensitivity does not necessarily leads to better finetuning performance across all blocks, although it is remarkably so for the feedforward normalization layer that is both highly sensitive and matches or surpasses full-finetuning while reducing the number of trainable parameters by a factor of 5000.

# 2 PRELIMINARIES

**Notations.**   The identity function over a Euclidean space is denoted by $\mathrm{id}\colon x \mapsto x$. The Euclidean norm on $\mathbb{R}^d$ is denoted by $\|\cdot\|$. The Frobenius inner product between matrices $\mathbf{A}, \mathbf{B} \in \mathbb{R}^{n \times m}$ is denoted by $\langle \mathbf{A}, \mathbf{B} \rangle = \sum_{i,j} \mathbf{A}_{ij} \mathbf{B}_{ij}$. The set of compactly supported probability measures on $\mathbb{R}^d$ is denoted by $\mathcal{P}_c(\mathbb{R}^d)$. The probability simplex is denoted by $\Delta_n := \{\mathbf{a} \in \mathbb{R}^n_+ \colon \sum_{i=1}^n \mathbf{a}_i = 1\}$.

**Transformers.** In this work, we focus on encoder-only transformers (Vaswani et al., 2017), which operate on sequences of tokens $x = (x_1, \ldots, x_n)$ of length $n$. The $x_i$ are first embedded in $\mathbb{R}^d$ and positional embeddings are then added to the tokens to capture the order in the sequence. The model is composed of a succession of transformer blocks, where each block applies a normalization layer, a multi-head self-attention block, a residual connection, followed by another normalization layer, a feedforward block, and a final residual connection. Note that the positional embedding is not part of the *transformer block*. The output of the transformer is a sequence of tokens in $\mathbb{R}^d$. The data modality and type of task will determine the embedding layer implementation and the task-specific layer applied on top of the transformer output. Let $E$ be the input space, $d$ be the embedding dimension, and $n$ be the sequence length. We denote by $f_{\mathrm{emb}} \colon E \to \mathbb{R}^d$ the embedding layer that embeds the tokens and adds the positional embedding, and by $f_{\mathrm{block}} \colon \mathbb{R}^d \to \mathbb{R}^d$ a transformer block, and by $f_{\mathrm{task}} \colon \mathbb{R}^d \to V$ the task-specific layer, with $V$ the output space (e.g., $\mathbb{R}^K$ with $K$ the number of classes). An encoder-only transformer $f_{\mathrm{enc}} \colon E \to V$ with $L$ layers can then be written as:

$$f_{\mathrm{enc}} = f_{\mathrm{task}} \circ \underbrace{f_{\mathrm{block}} \circ \cdots \circ f_{\mathrm{block}}}_{\times L} \circ f_{\mathrm{emb}}, \tag{1}$$

where each transformer block writes

$$f_{\mathrm{block}} = [\mathrm{id} + f_{\mathrm{ffn}} \circ f_{\mathrm{norm}}^{\mathrm{ffn}}] \circ [\mathrm{id} + f_{\mathrm{attn}} \circ f_{\mathrm{norm}}^{\mathrm{attn}}], \tag{2}$$

with $f_{\mathrm{norm}}^{\mathrm{attn}} \colon \mathbb{R}^d \to \mathbb{R}^d$ (and similarly $f_{\mathrm{norm}}^{\mathrm{ffn}}$) is a normalization layer, typically LayerNorm (Ba et al., 2016) or RMSNorm (Zhang & Sennrich, 2019), $f_{\mathrm{attn}} \colon \mathbb{R}^d \to \mathbb{R}^d$ the multi-head self-attention block (Vaswani et al., 2017), the residual connections are implemented through $\mathrm{id}$ and $f_{\mathrm{ffn}} \colon \mathbb{R}^d \to \mathbb{R}^d$ is the feedforward block. It is typically an MLP with two fully-connected layers $f_{\mathrm{fc1}} \colon \mathbb{R}^d \to \mathbb{R}^{4d}$ and $f_{\mathrm{fc2}} \colon \mathbb{R}^{4d} \to \mathbb{R}^d$ separated by a non-linear activation like ReLU (Agarap, 2018) or GeLU (Hendrycks & Gimpel, 2016). The implementation described above, which is the one used in modern models, slightly differs from the original one by placing the normalization layers before the attention and feedforward blocks, a method called pre-norm (Xiong et al., 2020). Other modifications are used in practice for efficiency, like flash-attention (Dao et al., 2022) or KV-caching (Pope et al., 2022), which do not fundamentally change the architecture described in Eq. (1) and Eq. (2).

**Distance between probability measures.** Optimal transport (OT) provides theoretical (Villani, 2008; Santambrogio, 2015) and computational (Peyré & Cuturi, 2019) tools to compare probability distributions while capturing the geometry of the underlying data. The 2-Wasserstein distance between $\mu, \nu \in \mathcal{P}_c(\mathbb{R}^d)$ can be defined as

$$\mathcal{W}_2(\mu, \nu) \coloneqq \left( \inf_{\pi \in \Pi(\mu,\nu)} \int \|x - y\|^2 \mathrm{d}\pi(x,y) \right)^{\frac{1}{2}}, \tag{3}$$

where $\Pi(\mu, \nu)$ is the set of couplings between $\mu$ and $\nu$, i.e., the probability measures on $\mathcal{P}_c(\mathbb{R}^d \times \mathbb{R}^d)$ satisfying $\int \pi(\cdot, y)\mathrm{d}y = \mu$ and $\int \pi(x, \cdot)\mathrm{d}x = \nu$ (see Santambrogio (2015) for more information on the subject). In the discrete case, which arises in many machine learning applications (Peyré & Cuturi, 2019; Liu et al., 2023; Flamary et al., 2021), the Wasserstein distance between empirical measures $\mu = \sum_{i=1}^n \mathbf{a}_i \delta_{x_i}$ and $\nu = \sum_{i=1}^m \mathbf{b}_j \delta_{y_j}$ amounts to solving the linear program: $\mathcal{W}_2(\mu, \nu) = \min_{\mathbf{P} \in \mathbf{U}(\mathbf{a},\mathbf{b})} \langle \mathbf{P}, \mathbf{C} \rangle$, where $\mathbf{U}(\mathbf{a}, \mathbf{b}) \coloneqq \{\mathbf{P} \in \mathbb{R}_+^{n \times m} : \mathbf{P}\mathbb{1}_n = \mathbf{a} \text{ and } \mathbf{P}^\top \mathbb{1}_m = \mathbf{b}\}$ and $\mathbf{C} \in \mathbb{R}_+^{n \times m}$ has entries $\|x_i - y_j\|^2$. In machine learning applications, the weights are often set as uniform $a_i = \frac{1}{n}$ and $b_i = \frac{1}{m}$.

## 3 TRANSFORMER SENSITIVITY TO DISTRIBUTION SHIFTS

Since foundation models can be applied to a wide range of downstream tasks and data modalities, it is important to study their sensitivity to distribution shifts at the level of the hidden representation. This, in turn, requires defining a notion of distance between sequences of tokens in order to quantify how outputs vary under input shifts. However, establishing such a distance is challenging: standard Euclidean tools are unreliable (Aggarwal et al., 2001) because embeddings are high-dimensional and prior similarity metrics between hidden representations—such as CKA (Kornblith et al., 2019)—have been shown to be sensitive to outliers and can yield counterintuitive

results (Davari et al., 2023). In this section, we show that any non-causal transformer can be seen as a succession of measure-to-measure maps by viewing sequences of tokens as discrete measures. This viewpoint allows us to define the sensitivity to distribution shift as an average notion of Lipschitz continuity, typically associated with robustness to perturbation, by comparing sequences of token with the Wasserstein distance, which captures the geometry of the underlying data.

## 3.1 DECOMPOSING TRANSFORMERS AS MEASURE-TO-MEASURE MAPS

This section introduces the decomposition of transformers into maps between discrete measures.

**Motivation.** Encoder-only transformers rely on the self-attention module $f_{\text{attn}}$ (Bahdanau et al., 2016), which is permutation equivariant (Castin et al., 2024). It means that for any input sequence of tokens $x = (x_1, \ldots, x_n) \in (\mathbb{R}^d)^n$ obtained after the embedding layer $f_{\text{emb}}$ and any permutation $\sigma$ of $\{1, \ldots, n\}$, we have

$$f_{\text{attn}}(x_{\sigma(1)}, \ldots, x_{\sigma(n)}) = (f_{\text{attn}}(x)_{\sigma(1)}, \ldots, f_{\text{attn}}(x)_{\sigma(n)}).$$

As such, $f_{\text{attn}}$ is blind to the order of the input tokens and can be equivalently seen as acting on the set $\{x_i\}_{i=1}^n$ or its associated empirical measure $\mu = \frac{1}{n} \sum_{i=1}^n \delta_{x_i} \in \mathcal{P}_c(\mathbb{R}^d)$ (Castin et al., 2024; Sander et al., 2022; De Bie et al., 2019). This leads to a natural formulation of $f_{\text{attn}}$ as a map between measures (Geshkovski et al., 2025), where we define the action of a map $T$ on $\mu$ through $T(\mu) = \frac{1}{n} \sum_{i=1}^n \delta_{T(x_i)}$.[1] This perspective has been used in the literature to theoretically study transformers, e.g., by extending the self-attention to continuous probability measure in the mean-field limit, i.e., with $n \to \infty$ (Sander et al., 2022; Castin et al., 2024), or to study the evolution of tokens across the layers as a dynamic system of interacting particles controlled by an ordinary differential equation (Geshkovski et al., 2025; Lu et al., 2019; Geshkovski et al., 2023). We note, however, that a formal mathematical treatment often requires one to simplify the architecture, for instance by considering self-attention only models. We extend this formulation to an entire transformer block in the next paragraph.

**Formulation.** Let $f_{\text{block}}$ be a transformer block. Using the fact that the attention module is the only one involving interactions between tokens, we obtain that all the trainable components of the transformer block, i.e., the $f_{\text{norm}}^{\text{attn}}$, $f_{\text{attn}}$, $f_{\text{norm}}^{\text{ffn}}$, $f_{\text{fc1}}$, and $f_{\text{fc2}}$ from Eq. (2), are permutation equivariant and thus naturally induce maps acting on discrete measures. Similarly to the attention described above, the action of a given component $f$ on $\mu$ writes $\frac{1}{n} \sum_{i=1}^n \delta_{f(x_i)}$ (by abuse of notation, we keep denoting by $f$ the associated map). It leads to the following decomposition of the transformer block:

$$\mu \in \mathcal{P}_c(\mathbb{R}^d) \xrightarrow{f_{\text{norm}}^{\text{attn}}} \mathcal{P}_c(\mathbb{R}^d) \xrightarrow{f_{\text{attn}}} \mathcal{P}_c(\mathbb{R}^d) \xrightarrow{f_{\text{norm}}^{\text{ffn}}} \mathcal{P}_c(\mathbb{R}^d) \xrightarrow{f_{\text{fc1}}} \mathcal{P}_c(\mathbb{R}^{4d}) \xrightarrow{f_{\text{fc2}}} \mathcal{P}_c(\mathbb{R}^d). \quad (4)$$

This formulation implies that the transformer block itself is a measure-to-measure map and so is the succession of transformer blocks. As shown in the following section, this perspective enables us to study how sensitive transformer components are to distribution shifts by working with a natural representation of sequences of tokens and defining a meaningful distance.

**Remark 3.1.** *We note that the residual connections do not impact Eq. (4), they only act on the locations of the measures to be fed to the map after them. Since our focus is on the trainable components sensitive to shifts, we do not make them appear in our formulation.*

## 3.2 SENSITIVITY TO DISTRIBUTION SHIFTS

In this section, we leverage the previous formulation to define the sensitivity to shifts as a notion of Lipschitz continuity.

**Intuition.** Consider a dataset $\mathcal{D}$ with the same data distribution as to pretraining data and a dataset $\mathcal{D}_{\text{ood}}$ representing the downstream task, with the assumption that it corresponds to out-of-distribution data. We consider two input sequences $(x_1, \ldots, x_n) \in (\mathbb{R}^d)^n$ and $(y_1, \ldots, y_n) \in$

---

[1]This amounts to the push-forward operator of $T$ in the discrete case (Sander et al., 2022; Peyré & Cuturi, 2019).

$(\mathbb{R}^d)^n$ sampled from $\mathcal{D}$ and $\mathcal{D}_{\text{ood}}$ respectively, and denote by $\mu$, $\nu$ their associated discrete measures. We conjecture that sensitive layers $f$ might amplify the distance between $\mathcal{W}_2(\mu, \nu)$ and $\mathcal{W}_2(f(\mu), f(\nu))$ while robust ones would preserve it. Lipschitz continuity provides a classical way of controlling the regularity of a function, studying how fast its output varies with respect to its input and commonly associated to robustness (Rosca et al., 2020; Goodfellow et al., 2015; Miyato et al., 2016; Tsuzuku et al., 2018; Gao et al., 2023; Rosca et al., 2020). It has notably been used to study the regularity of self-attention (see Castin et al., 2024).

**Approach.** While (local) Lipschitz constants are typically formulated as the maximum rate of change of outputs with respect to inputs, this quantity is highly sensitive to outliers, reflecting a worst case behavior, and fails to capture an *overall* shift between distributions. This motivates us to define the sensitivity to shift as an average notion of Lipschitz continuity:

> **Definition 3.1** (Sensitivity to distribution shifts). *Let $\mathcal{E}(\mathcal{D})$ be the set of sequences of tokens obtained after embedding the data from $\mathcal{D}$. We define $\mathcal{E}(\mathcal{D}_{\text{ood}})$ similarly. Consider a transformer component $f \colon \mathcal{P}_c(\mathbb{R}^m) \to \mathcal{P}_c(\mathbb{R}^p)$ seen as a measure-to-measure map. We define its sensitivity to the distribution shift induced by $\mathcal{D}_{\text{ood}}$ as*
>
> $$\mathcal{S}(f, \mathcal{D}, \mathcal{D}_{\text{ood}}) := \mathbb{E}_{(\mu,\nu) \in \mathcal{E}(\mathcal{D}, \mathcal{D}_{\text{ood}})} \left[ \frac{\mathcal{W}_2(f(\mu), f(\nu))}{\mathcal{W}_2(\mu, \nu)} \right], \qquad (5)$$
>
> *where $\mathcal{E}(\mathcal{D}, \mathcal{D}_{\text{ood}})$ is the set of pairs of distinct discrete measures $(\mu, \nu)$ associated to the sequences of tokens from $\mathcal{D}$ and $\mathcal{D}_{\text{ood}}$ obtained after embedding. When clear from the context, we denote it $\mathcal{S}(f)$ for short.*

The metric properties of $\mathcal{W}_2$ are instrumental in ensuring that $\mathcal{S}(f)$ is well defined for distinct probability measures. While unbounded in principle, we have $\mathcal{S}(f)$ nonnegative and equal to zero almost everywhere if and only if $f$ maps all the input sequences of tokens to the same measure. This would typically be the case of a function whose output is independent of the values of the input. The case where $\mathcal{S}(f) = 1$ indicates that overall, the map $f$ preserves the original distance between $\mathcal{D}$ and $\mathcal{D}_{\text{ood}}$, and can thus be coined as neutral to the shift. Similarly, $\mathcal{S}(f) > 1$ would indicate a map amplifying the shift, while $\mathcal{S}(f) < 1$ would indicate a map that attenuates it. Since transformer models consist of a succession of similar blocks Eq. (1), one can recover the average sensitivity of a component $f$ over the depth to obtain global information, as shown in Section 4.2.

**Practical implementation.** To conduct experiments with large models, we adopt an efficient implementation of the sensitivity relying on tensor computation (Paszke et al., 2019). For a given component $f$, we loop over batches of data and compute the exact Wasserstein distance using the POT library (Flamary et al., 2021). The main computational burder to compute the Wasserstein distance is the sequence length $n$. In our experiments, we can scale to large vision transformers (up to $632M$) with the exact Wasserstein using the fact that patched images lead to reasonable values of $n$. For large language models, the sequence length is often bigger (for instance, Gemini 2.5 (Comanici et al., 2025) has a context window of 1 million), and efficient approximate of the Wasserstein might be preferred (Cuturi, 2013; Bonneel et al., 2015; Peyré & Cuturi, 2019).

## 4 EXPERIMENTS

In this section, we study the sensitivity to shifts of large vision transformers (Dosovitskiy et al., 2021) pretrained on ImageNet-21k (Deng et al., 2009), the superset of the ILSVRC-2012 ImageNet dataset with 21k classes and 14M images. We then investigate how sensitivity translates to potential gains when finetuning the components in isolation.

### 4.1 EXPERIMENTAL SETUP

**Datasets.** We consider open-source benchmarks commonly used in the image classification literature. The sensitivity analysis is conducted on ImageNet (Deng et al., 2009) and ImageNet-C (Hendrycks & Dietterich, 2019), which consists of corruptions of the original ImageNet. Finetun-

ing is performed on Cifar10 & Cifar100 (Krizhevsky, 2009), Cifar10-C (Hendrycks & Dietterich, 2019) which consists of corrupted images from Cifar10, Oxford-IIIT Pets (Parkhi et al., 2012) and Oxford Flowers-102 (Nilsback & Zisserman, 2008). For preprocessing, we follow Kolesnikov et al. (2020) and apply random cropping followed by a 224×224 image resizing and random horizontal flip for training images. For validation and test data, the 224×224 image resizing is applied before center cropping images. All images are normalized using the ImageNet (Deng et al., 2009) statistics[2].

**Models.** Vision transformers (ViT) follow the architecture described in Eq. (1). Images are first embedded in dimension $d$ by splitting them into smaller patches, which play the role of tokens. These patches are flattened and projected into $\mathbb{R}^d$ with a linear layer. A positional embedding layer is then applied to retain the positional implementation. The token embedding and positional embedding can also be done at once using a convolutional layer (see Dosovitskiy et al., 2021, Section 3.1). An additional classification token CLS is prepended to the embedded sequence and fed to the classification head $f_{\text{task}}$ for the final prediction. The ViT models come in three flavors (Base, Large, Huge) with patch sizes 16, 16, and 14, respectively. Details on the models' variants are given in Table 3 along with our implementation in Fig. 4.

**Training details.** We follow the finetuning setup from Dosovitskiy et al. (2021): optimization is done with Stochastic Gradient Descent (SGD) with a momentum of 0.9 and no weight decay, a cosine learning rate decay, a batch size of 512, and gradient clipping at norm 1. The finetuning resolution is of 224 and for each dataset, we make a sweep over 4 learning rates, as summarized in Table 4. More details on our finetuning setup are provided in Appendix A.2.

### 4.2 WHICH TRANSFORMER COMPONENTS ARE THE MOST SENSITIVE TO SHIFT?

We begin by investigating the sensitivity to distribution shifts of ViT models. Following Definition 3.1, we set $\mathcal{D}$ and $\mathcal{D}_{\text{ood}}$ as the validation sets of ImageNet and ImageNet-C, respectively. To ensure diverse shifts, we consider 6 types of corruption (brightness, contrast, gaussian noise, motion blur, snow, speckle noise) with 5 levels of severity for a total of 30 corrupted versions of ImageNet. For each component $f$, we compute its sensitivity $\mathcal{S}(f)$ averaged over the layers.

**Findings.** We display the results with ViT-Base (top) and ViT-Large (bottom) for each pair (ImageNet, ImageNet-C) on the gaussian noise, motion blur, and snow corruptions for all the severities in Fig. 2. The different components are indicated with markers of colors (see Table 5 for the list of components) and the neutral scenario $\mathcal{S}(f) = 1$ is indicated by the dashed red line. We observe a consistent pattern over datasets and models, with the attention and feedforward normalization layers being the most sensitive ones. We note that the attention block is sensitive to shift for ViT-Base, although less than the normalization norms, while the feedforward components are below or on par with the line $\mathcal{S}(f) = 1$, indicating that those layers are robust to distribution shifts. The gap between the attention and feedforward blocks closes for the ViT-Large while the normalization layers remain consistently sensitive. In Appendix A.3, we display the results for the same experiment with ViT-Huge and with 3 additional corruptions (brightness, contrast, speckle noise) on ViT-Base and observe a similar behavior. Fig. 1 (*Left*) summarizes these global patterns by averaging over the 30 corrupted versions of ImageNet.

> **Takeaway 1.** Normalization layers are consistently the most sensitive to distribution shifts across datasets and model sizes. They are followed by the attention modules and feedforward layers.

### 4.3 DOES THE SENSITIVITY TO SHIFT HELP WHEN ADAPTING TO NEW DATA?

The findings from the previous section indicate that the normalizing layers are the most sensitive components, in the sense that they tend to stretch the distance (see Definition 3.1) between sequences of tokens coming from different distributions. A natural question to ask is whether this flexibility can lead to an easier adaptation to new data. In this section, we investigate this question by evaluating the finetuning performance of each ViT component in isolation.

---

[2]It ensures images with mean $[0.485, 0.456, 0.406]$ and standard deviation $[0.229, 0.224, 0.225]$.

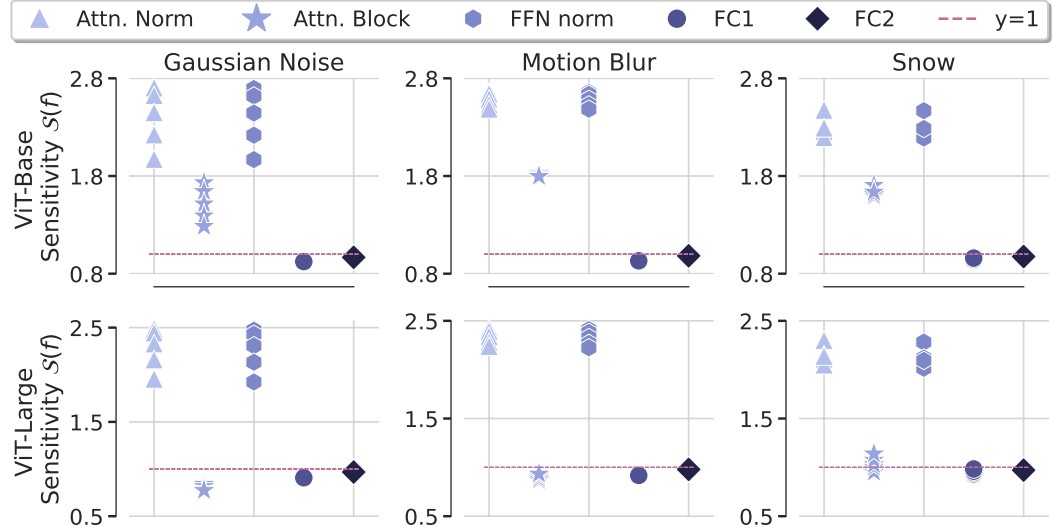

Figure 2: **Sensitivity to distribution shifts.** For each component $f$ of ViT-Base (top) and ViT-Large (bottom), we display the sensitivity $\mathcal{S}(f)$, defined in Definition 3.1, averaged over the depth, for each pair (ImageNet, ImageNet-C) on the gaussian noise, motion blur and snow corruptions with severity in $\{1, 2, 3, 4, 5\}$. Components are indicated by markers of colors (see Table 5 for the list of components) and the neutral scenario $\mathcal{S}(f) = 1$ is indicated by the dashed red line.

**Setting.** We now finetune a pretrained model on a downstream task with the aim to optimize both the performance and the computational cost (e.g., training memory usage). While in some scenarios, it can be useful to classify distribution shifts into categories (Xie et al., 2024; Lee et al., 2023), our finetuning experiments involve a shift between the pretraining data (ImageNet-21k) and other common vision benchmarks for which the exact type of shift is not explicitly known.

**Finetuning configurations.** We consider the ViT-Base model and to study how the sensitivity to distribution shifts impacts the adaptation to new data, we freeze all the trainable components of the model, except the one studied, among the attention norm, the attention module, the feedforward norm, the first feedforward layer and the second feedforward layer across the blocks, and add a randomly initialized linear layer for the classification following Dosovitskiy et al. (2021). We add the full-finetuning where all the model's parameters are trainable as a baseline, which leads to 6 finetuning configurations, described in Table 5 along with their corresponding number of trainable parameters. When specified, seeds are relative to dataloaders and network initialization.

**Performance comparison.** We display the results obtained for 3 finetuning runs in Table 1 of the ViT-Base over Cifar10, Cifar100, Cifar10 with gaussian noise and motion blur corruptions with a severity of 5, Oxford-IIIT Pet and Oxford Flowers-102 datasets. The same experiment is displayed in Fig. 3 with relative performance compared to full-finetuning. Our immediate conclusion is that the sensitivity in the sense of Lipschitz continuity does not necessarily correlate with the downstream finetuning performance across blocks. This becomes obvious when comparing the superior performance of the finetuning feedforward layers (least sensitive) with that of the LayerNorms (most sensitive). A more nuanced analysis reveals that finetuning the first feedforward layer is consistently better than the second one. Similarly, finetuning the pre-feedforward normalization layer is more efficient than finetuning the pre-attention one. Finally, finetuning the attention block remains the closest alternative in terms of efficiency to finetuning the first feedforward layer. Fig. 1 (*Right*) summarizes these global patterns by averaging over the benchmarks.These findings indicate that measuring the sensitivity of the transformer blocks is not universally informative across all layers, yet it correctly identifies the sensitivity of the normalization layers and the potential gains of finetuning them may bring.

Table 1: Performance comparison between finetuning configurations. The metric reported is the top-1 accuracy on the test set of each benchmark (↑). Gaussian Noise and Motion Blur corresponds to the gaussian noise and motion blur corrupted version of CIFAR10-C with severity 5. The entries show the mean and standard deviation over three finetuning runs. **Best** results are in bold, second best are underlined.

| Dataset | ViT trainable components | | | | | |
| --- | --- | --- | --- | --- | --- | --- |
| | Attn. Norm | Attn. Block | FFN Norm | FC1 | FC2 | All |
| Cifar10 | $98.67_{\pm 0.03}$ | $98.96_{\pm 0.00}$ | $98.72_{\pm 0.05}$ | $\mathbf{99.09}_{\pm 0.05}$ | $98.91_{\pm 0.06}$ | $\underline{99.04}_{\pm 0.01}$ |
| Cifar100 | $91.43_{\pm 0.07}$ | $92.65_{\pm 0.07}$ | $91.93_{\pm 0.11}$ | $\mathbf{92.85}_{\pm 0.07}$ | $92.34_{\pm 0.07}$ | $\underline{92.74}_{\pm 0.05}$ |
| Gaussian Noise | $88.99_{\pm 0.24}$ | $89.41_{\pm 0.53}$ | $\mathbf{89.55}_{\pm 0.04}$ | $\underline{89.49}_{\pm 0.16}$ | $88.49_{\pm 0.51}$ | $87.87_{\pm 1.07}$ |
| Motion Blur | $93.29_{\pm 0.24}$ | $\underline{94.75}_{\pm 0.25}$ | $94.10_{\pm 0.19}$ | $94.57_{\pm 0.05}$ | $94.05_{\pm 0.15}$ | $\mathbf{94.67}_{\pm 0.14}$ |
| Oxford-IIIT Pets | $\underline{94.47}_{\pm 0.08}$ | $94.42_{\pm 0.06}$ | $\underline{94.47}_{\pm 0.11}$ | $94.27_{\pm 0.25}$ | $93.98_{\pm 0.20}$ | $\mathbf{94.63}_{\pm 0.21}$ |
| Oxford Flowers-102 | $98.92_{\pm 0.21}$ | $98.97_{\pm 0.02}$ | $\mathbf{99.19}_{\pm 0.06}$ | $99.01_{\pm 0.0}$ | $98.83_{\pm 0.05}$ | $\underline{99.18}_{\pm 0.02}$ |

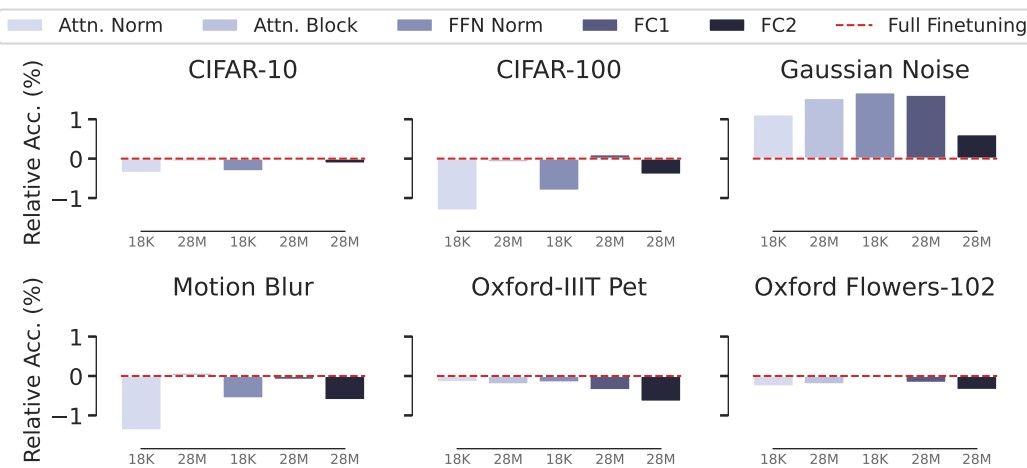

Figure 3: **Benefits of finetuning sensitive components**. We display the relative accuracy (difference between the performance of finetuning a component in isolation and the full-finetuning performance) of the components of a pretrained ViT-Base model, over the 6 benchmarks from in Table 1. The number of trainable parameters of each component is displayed in the x-axis (see Table 5). The red dashed line indicates the level of 0, which represents the full-finetuning performance.

**Signal propagation.** We note that the analysis of Section 4.2 does not take into account the position of a given component in a transformer block. This could explain the different behavior observed between the feedforward blocks. Indeed, the first layer FC1 is followed by a non-linear activation $\sigma$, which is not the case of the second one, FC2. As such, finetuning FC1 can be seen equivalently as tuning a non-linear layer, providing more expressivity.

**Memory usage.** Since the different model configurations considered in our finetuning experiments share the same architecture, their inference cost is the same. However, the change in terms of the number of trainable parameters impacts the cost of training them. In terms of VRAM, the GPU usage of training a model consists of the memory load to store the model parameters, the optimizer states, the gradients, and the activations (Thor, 2025). In our case, the memory load for the different configurations will only differ in terms of optimizer and gradient computation. For a model with $P$ parameters and a precision of $b$ bytes, the memory required to store gradients amounts to $Pb$ because backpropagation computes a gradient for each parameter. The same memory is needed

Table 2: Rank of each configuration and memory usage per training step.

| Configuration | Rank | Memory (MB) |
| --- | --- | --- |
| Attn. Norm | 5 | 0.14 |
| Attn. Block | 1 | 220 |
| FFN Norm | 4 | 0.14 |
| FC1 | 2 | 220 |
| FC2 | 6 | 220 |
| All | 3 | 690 |

for the optimizer states with SGD (and the double for Adam (Kingma & Ba, 2014; Loshchilov & Hutter, 2019), which also computes the variance). In Table 2, we display the overall ranking of each configuration in terms of finetuning performance along with the memory load to compute gradients and store the optimizer with the default FP32 precision for one training step. We observe that the normalization layers are competitive while reducing the memory load by $\sim 5000$ compared to full-finetuning.

> **Takeaway 2.** Increased sensitivity does not universally predict finetuning performance across all blocks. Yet, it is remarkably the case for the feedforward normalization layer that is both highly sensitive and leads to competitive results after finetuning.

## 5 RELATED WORK

**Neural networks acting on measures.** The study of neural networks acting on measures has been first proposed in (De Bie et al., 2019; Pevny & Kovarik, 2019). It has been mostly used to conduct theoretical studies on transformers, e.g., to determine the Lipschitz constant of attention (Castin et al., 2024) or offer a mathematical perspective of transformers as interacting particles (Geshkovski et al., 2023; 2025; Lu et al., 2019). In Sander et al. (2022), the authors used this viewpoint to propose a novel normalization of attention in transformers and conduct a theoretical analysis of it. In our work, we extend this viewpoint to the entire architecture, which enables us to use to conduct large-scale experiments on pretrained transformers.

**Finetuning foundation models.** With the increasing scale of foundation models, a lot of effort has been put into reducing the computational cost of finetuning models (HuggingFace, 2025). (Hu et al., 2022) replaces the weights by trainable rank decomposition matrices, and (Ye et al., 2023) proposes an adaptive selection of attention and feedforward layers, thus reducing the number of trainable parameters for the downstream task. In our work, we study each transformer component in isolation since our aim is mainly to better understand the architecture in terms of distribution shifts. Our approach can be combined with parameter-efficient finetuning like LoRa by applying it only for the components sensitive to shifts, further reducing the cost of finetuning. Another related work is (Lee et al., 2023), which studies which layers of neural networks are the best to finetune depending on specific types of shifts. In our work, we focus on the transformer model in a component-wise manner, without assumptions on the shift occurring between pretraining and downstream data.

## 6 DISCUSSION

In our work, we propose an approach to study the sensitivity to distribution shifts of pretrained transformer models. Decomposing them into measure-to-measure maps allows us to compare sequences of tokens with a meaningful notion of distance, which we use to define the sensitivity of transformer components as an average Lipschitz continuity notion. Through comprehensive experiments on large vision transformers, we identify that the normalization layers are consistently the most sensitive parts. While we do not observe that increased sensitivity steadily leads to better finetuning performance across all blocks, it is remarkably the case for the feedforward normalization layer that is both highly sensitive and matches or surpasses full-finetuning while reducing the number of trainable parameters by a factor of 5000. Overall, our findings provide new insights into how transformer components behave under distribution shifts, showcasing that a better understanding of the transformer architecture can inform the design of more efficient adaptation methods.

**Limitations and future work.** Although we concentrate primarily on computer vision datasets in our work due to the availability of common OOD pairs for finetuning (ImageNet to CIFAR, etc.), our analysis can be extended to other domains such as NLP and time series. On a somewhat different note, our work provides a comprehensive empirical evaluation of the usefulness of Lipschitz continuity in predicting the downstream finetuning performance, but it does not provide counterexamples explaining the mismatch between the empirical results and the theoretical intuitions. Finally, to carefully disentangle the measured sensitivity and its impact on performance, we do not perform a thorough investigation into the most recent finetuning procedures, which come with algorithmic heuristics.

ETHICS STATEMENT

This paper presents work whose goal is to advance the field of Machine Learning. There are many potential societal consequences of our work, none of which we feel must be specifically highlighted here.

REPRODUCIBILITY STATEMENT

All the details to reproduce our results are provided in the main paper and the appendix. We used open-source models and datasets described in Section 4.1 and provide the implementation details along with the experimental setups in Section 4.2 and Section 4.3. Our code will be open-sourced upon publication.

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

# Appendix

## A  EXPERIMENTS

### A.1  VISION TRANSFORMER IMPLEMENTATION

In our code, we follow the original ViT implementation from Dosovitskiy et al. (2021) and use a convolutional layer to embed images (see Dosovitskiy et al., 2021, §"Hybrid Architecture"). In Fig. 4, we display the implementation of the ViT-Base model with a classification head for 10 classes (we renamed our package "my_lib" to respect the anonymity). The different models' variants are described in Table 3 along with the weights from HuggingFace.

```python
# Python snippet to show the ViT architecture
from my_lib import ViT

model = ViT(name="base", n_classes=10)
print(model)

# Corresponding output
Transformer(
  (embedding): Embedding(
    (patching): PatchImages(
      (patching): Sequential(
        (0): Conv2d(3, 768, kernel_size=(16, 16), stride=(16, 16))
        (1): Flatten(start_dim=2, end_dim=-1)
      )
    )
  )
  (blocks): ModuleList(
    (0-11): 12 x TransformerBlock(
      (attn_norm): LayerNorm((768,), eps=1e-12, elementwise_affine=
          True)
      (attn): SelfAttention(
        (qkv_mat): Linear(in_features=768, out_features=2304, bias=
            True)
        (output): Linear(in_features=768, out_features=768, bias=True
            )
      )
      (ffn_norm): LayerNorm((768,), eps=1e-12, elementwise_affine=
          True)
      (ffn): FeedForward(
        (fc1): Linear(in_features=768, out_features=3072, bias=True)
        (fc2): Linear(in_features=3072, out_features=768, bias=True)
      )
    )
  )
  (output): Output(
    (output_layer): ClassificationLayer(
      (output_norm): LayerNorm((768,), eps=1e-12, elementwise_affine=
          True)
      (output): Linear(in_features=768, out_features=10, bias=True)
    )
  )
)
```

Figure 4: ViT-Base Implementation.

Table 3: Details of ViT variants (Dosovitskiy et al., 2021) with the number of layers, number of attention heads, embedding dimension, the number of parameters, and the pretrained weights.

| Model | Layers | Heads | Embedding | Params | Weights |
|-------|--------|-------|-----------|--------|---------|
| ViT-Base | 12 | 12 | 768 | 86M | vit-base-patch16-224 |
| ViT-Large | 24 | 16 | 1024 | 307M | vit-large-patch16-224 |
| ViT-Huge | 32 | 16 | 1280 | 632M | vit-huge-patch14-224 |

## A.2 TRAINING SETUP

Our finetuning setup is borrowed from Dosovitskiy et al. (2021). We optimize models with the Stochastic Gradient Descent (SGD), a momentum of 0.9, and no weight decay, a cosine learning rate decay, a batch size of 512, and gradient clipping at norm 1. The finetuning resolution is of 224 and for each dataset, we make a sweep over 4 learning rates, as summarized in Table 4, and report the best performance obtained. Since Cifar10-C corresponds to the validation set of Cifar10, we manually create deterministic training and test sets following a $80\% - 20\%$ split. For all datasets, we monitor training with a validation set ($20\%$ of the training set). The final performance is the test accuracy of the model that achieves the best validation accuracy. For each model and set of hyperparameters, we conduct 3 runs (2 on Oxford Flowers-102) and report the corresponding mean and standard deviation. In Table 5, we display the different finetuning configurations along with their number of trainable parameters.

Table 4: Hyperparameters for finetuning.

| Dataset | Training steps | Base learning rates |
|---------|----------------|---------------------|
| Cifar10 | 10 000 | $\{0.001, 0.003, 0.01, 0.03\}$ |
| Cifar100 | 10 000 | $\{0.001, 0.003, 0.01, 0.03\}$ |
| Cifar10-C | 10 000 | $\{0.001, 0.003, 0.01, 0.03\}$ |
| DomainNet | 20 000 | $\{0.003, 0.01, 0.03, 0.06\}$ |
| Oxford-IIIT Pets | 4 000 | $\{0.001, 0.003, 0.01, 0.03\}$ |
| Oxford Flowers-102 | 5 000 | $\{0.001, 0.003, 0.01, 0.03\}$ |

Table 5: Trainable components of transformers following Eq. (1) and Eq. (2) and their corresponding finetuning configuration with the number of trainable parameters.

| Configuration
Component $f$ | Attn. Norm
$f_{\text{norm}}^{\text{attn}}$ | Attn. Block
$f_{\text{attn}}$ | FFN Norm
$f_{\text{norm}}^{\text{ffn}}$ | FC1
$f_{\text{fc1}}$ | FC2
$f_{\text{fc2}}$ | All
$f_{\text{enc}}$ |
|---|---|---|---|---|---|---|
| Params | 18K | 28M | 18K | 28M | 28M | 86M |
| % of total | 0.02% | 33% | 0.02% | 33% | 33% | - |

## A.3 ADDITIONAL EXPERIMENTS

We display below additional experiments relative to the sensitivity analysis from Section 4.2, extending it on ViT-Huge and on three additional corruptions for ViT-Base. Our findings are consistent with those from Section 4.2 over the 30 corrupted variants of ImageNet, as can be seen in Fig. 5, with the attention and feedforward normalization layers being the most sensitive ones. In Fig. 6, we observe that with the ViT-Huge, although the normalization layers remain the most sensitive components over datasets, we observe that the attention block is more sensitive than what it was with ViT-Base and ViT-Large, while the behavior of the other components remain the same.

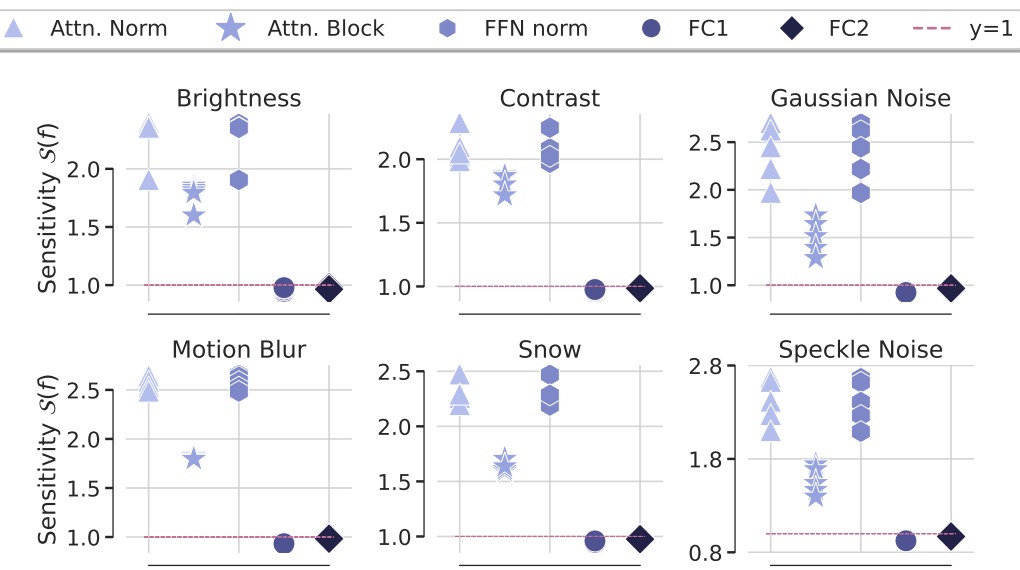

Figure 5: **Sensitivity to distribution shifts.** Similar experiment than Fig. 5 with ViT-Base on the brightness, contrast, and speckle noise with severity in $\{1, 2, 3, 4, 5\}$. Components are indicated by markers of colors (see Table 5 for the list of components) and the neutral scenario $\mathcal{S}(f) = 1$ is indicated by the dashed red line.

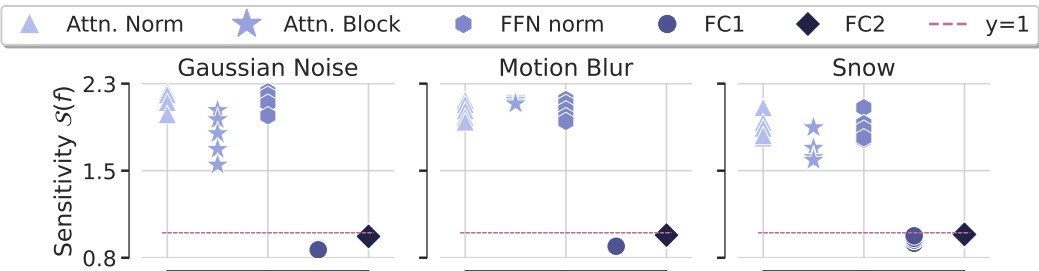

Figure 6: **Sensitivity to distribution shifts.** Similar experiment than Fig. 5 with ViT-Huge on the gaussian noise, motion blur, and snow corruptions with severity in $\{1, 2, 3, 4, 5\}$. Components are indicated by markers of colors (see Table 5 for the list of components) and the neutral scenario $\mathcal{S}(f) = 1$ is indicated by the dashed red line.

