# OpenReview forum: "Which transformers components are the most sensitive to distribution shifts?"
_ICLR.cc/2026/Conference — Submitted to ICLR 2026_

### Official Review · Reviewer_BXdx · 2025-10-26

**Soundness:** 2
**Presentation:** 2
**Contribution:** 1
**Rating:** 2
**Confidence:** 3

**Summary:**

This paper proposes a method to measure the sensitivity of Transformer components to distribution shifts. Through sensitivity analysis on ImageNet-C and fine-tuning performances on downstream tasks with vision Transformers, the paper finds that the FFN normalization layers are both highly sensitive and deliver strong fine-tuning performance.

**Strengths:**

1. Introduces a method to quantify component sensitivity using a notion of Lipschitz continuity.

2. Conducts extensive experiments across three ViT variants and multiple fine-tuning tasks, providing some empirical insights.

**Weaknesses:**

1.  The literature review overlooks several existing works investigating the importance of different Transformer components [1-5]. Notably, [1] has already empirically demonstrated the critical role of normalization layers, with their Tab. 6 specifically highlighting the significance of the FFN normalization layer.

2. The proposed sensitivity metric lacks convincing empirical validation, which limits its practical utility. Although the paper attempts to connect component sensitivity to fine-tuning performance, the correlation appears weak.

3. The sensitivity analysis is conducted primarily on the synthetic corruptions on ImageNet-C. More evaluation of natural distribution shifts could be helpful.

[1] Scaling & Shifting Your Features: A New Baseline for Efficient Model Tuning, NeurIPS 2022

[2] Locating and Editing Factual Associations in GPT, NeurIPS 2022

[3] Sensitivity-Aware Visual Parameter-Efficient Fine-Tuning, ICCV 2023

[4] Understanding and Patching Compositional Multihop Reasoning in Large Language Models, ACL Findings 2024

[5] Decomposing and Editing Predictions by Modeling Model Computation, ICML 2024

**Questions:**

1. For the sensitivity calculation for each corruption, is the metric computed between a natural image and its corresponding corrupted version, or a random image from the same corruption?

2. Could the paper show layer-wise sensitivity analysis to reveal how each component's sensitivity evolves across different layers?

---

### Official Review · Reviewer_fsTT · 2025-10-30

**Soundness:** 3
**Presentation:** 3
**Contribution:** 2
**Rating:** 4
**Confidence:** 4

**Summary:**

The paper studies robustness of Vision Transformers (ViTs) under distributional shift by modeling each component as a mapping between probability measures and evaluating its average Lipschitz continuity using the Wasserstein-2 distance. This metric measures how much a component amplifies differences between input distributions, providing a geometric notion of sensitivity. Experiments on ImageNet-C and several downstream datasets show that normalization layers are sensitive under this measure, and that fine-tuning only these layers can recover most of the performance of full fine-tuning.

**Strengths:**

1. The component-wise analysis of Transformer is informative, offering insights into where robustness degradation occurs (e.g., attention vs. normalization vs. feedforward blocks).
2. The paper presents a mathematically grounded framework for analyzing Transformer robustness through a measure-to-measure mapping view, connecting Lipschitz continuity with the Wasserstein-2 distance. This extends prior work on average sensitivity [1] by capturing global, distribution-level perturbations.
3. The presentation and organization are clear, and the experiments are across multiple datasets and model scales.

**Weaknesses:**

1. While the use of $W_2$ is theoretically well-motivated for comparing probability measures, the experimental setup (ImageNet-C corruptions) does not alter token order. In this regime, $W_2$ effectively reduces to an averaged $L_2$ distance. The paper would be stronger by including experiments involving token rearrangement that alter token structure (e.g., OOD language tasks [2] or Machine Translation [3] where word order and structure differ substantially across languages).
2. The conclusion that fine-tuning normalization layers matches full fine-tuning seems to rely on adapting directly to the OOD datasets (i.e., the shifted distributions themselves). If seen as a form of test-time adaptation of normalization statistics, this result corresponds to the known technique and insight of Tent [4], which also adapts normalization parameters via entropy minimization. Moreover, this setup no longer evaluates true OOD generalization. In realistic scenarios, such as adapting to a specialized domain or new task, the paper does not discuss whether tuning only normalization layers would still be effective.
3. The improvements reported in Table 1 are small, with most components achieving similar performance. As a result, the empirical conclusions about “most effective” components are not convincing.

**Questions:**

1. Could the authors clarify whether $W_2$ provides qualitatively different sensitivity rankings compared to using $L_2$ directly? This would help assess whether the Wasserstein geometry contributes distinct insights in this experimental regime.
2. Could the authors discuss whether fine-tuning only normalization layers would be effective when adapting to new domains or tasks that introduce genuinely new information, rather than merely distributional shifts?
3. Why does the measured component sensitivity not directly correspond to fine-tuning performance?

[1] Michael Hahn and Mark Rofin. Why are Sensitive Functions Hard for Transformers? ACL 2024.

[2] Di Jin, Zhijing Jin, Joey Tianyi Zhou, and Peter Szolovits. Is BERT Really Robust? A Strong Baseline for Natural Language Attack on Text Classification and Entailment. AAAI 2020.

[3] Alexis Conneau, Guillaume Lample, Ruty Rinott, Adina Williams, Samuel R. Bowman, Holger Schwenk, and Veselin Stoyanov. XNLI: Evaluating Cross-lingual Sentence Representations. EMNLP 2018.

[4] Dequan Wang, Evan Shelhamer, Shaoteng Liu, Bruno Olshausen, and Trevor Darrell. Tent: Fully Test-time Adaptation by Entropy Minimization. ICLR 2021.

---

### Official Review · Reviewer_9KFp · 2025-11-01

**Soundness:** 2
**Presentation:** 2
**Contribution:** 2
**Rating:** 2
**Confidence:** 4

**Summary:**

The paper proposes a novel sensitivity measure for transformer components under distribution shifts by formulating transformers as measure-to-measure maps and leveraging the notion of Lipschitz continuity. Through comprehensive experiments on large vision transformers across multiple corrupted datasets, the authors empirically investigate which components are most sensitive to distribution shifts during fine-tuning. Their key findings reveal that 1) normalization layers consistently exhibit the highest sensitivity, 2) increased sensitivity does not uniformly correlate with improved fine-tuning performance across all blocks, and 3) the feedforward normalization layer demonstrates both high sensitivity and achieves performance comparable to or exceeding full fine-tuning.

**Strengths:**

- The paper evaluates fine-tuning performance across at least 30 corrupted settings, providing substantial experimental evidence.
- The writing is easy to follow, with the main findings effectively communicated to readers.

**Weaknesses:**

- The paper lacks critical implementation details regarding the proposed sensitivity measure (see questions below for specifics).
- A significant limitation is the absence of comparative analysis with alternative sensitivity measures, such as CKA or standard Lipschitz continuity metrics. Understanding how these established methods would perform and whether they yield similar findings is essential, particularly given the authors' claim (lines 76-77) that prior methods are either unreliable or sensitive to outliers. Without such comparison, it is difficult to assess what new insights the proposed measure provides beyond existing approaches.

**Questions:**

- Please provide a more detailed explanation of how Equation 5 is computed in practice, as this represents the core contribution of the proposed measure. Additionally, clarify the computational complexity associated with computing this measure, as the current presentation glosses over these important details.
- In Figure 2 (sensitivity to distribution shifts), the correspondence between sensitivity values and corruption severity levels is difficult to discern (as all dots have the same colour). Is there a discernible pattern that emerges? Please clarify this relationship.
- The sensitivity patterns across different experiments in Figure 2 appear remarkably consistent, even across different corruption types. Could the authors provide an explanation for this consistency? Is there supporting evidence or theoretical justification for this observation? Alternatively, could this consistency arise from implicit biases in the sensitivity measure design that inherently favor specific blocks or layers within transformers? Additional details on the exact computation of sensitivity metrics would help address this concern.
- Given that the authors acknowledge higher sensitivity (as measured by their proposed metric) does not consistently lead to improved fine-tuning performance, what practical insights or benefits does this sensitivity measure provide immediately? Furthermore, prior work [1,2] has also highlighted the benefits of fine-tuning normalization layers or selected blocks in transformers. Could the authors discuss how their findings relate to or differ from these existing results?

[1] Basu et al., Strong Baselines for Parameter-Efficient Few-Shot Fine-Tuning, AAAI 2024
[2] Chen et al., Unleashing the Power of Meta-tuning for Few-shot Generalization Through Sparse Interpolated Experts, ICML 2024

---

### Official Review · Reviewer_np4Z · 2025-11-02

**Soundness:** 3
**Presentation:** 3
**Contribution:** 2
**Rating:** 4
**Confidence:** 2

**Summary:**

This submission studies which parts of a transformer are most sensitive to distribution shift and whether that sensitivity should guide parameter‑efficient fine‑tuning. The authors model each submodule—attention normalization, self‑attention, MLP normalization ("FFN‑Norm"), and the MLP’s two linear layers ("FC1" and "FC2")—as a measure‑to‑measure map acting on token embeddings. They then define a component‑level sensitivity score using optimal transport:

$$S(f; D, D_{\text{ood}}) = \mathbb{E} [\frac{ W_2\big(f(\mu), f(\nu)\big) }{ W_2(\mu, \nu) }] . $$

Here, $S\approx 1$ suggests the component is shift‑neutral, $S>1$ amplifies shift, and $S<1$ attenuates it. Experiments focus on Vision Transformers (ViTs), where exact $W_2$ is computationally feasible because sequences are patch‑length. Sensitivity is measured with ImageNet as in‑distribution and ImageNet‑C (multiple corruption types and severities) as out‑of‑distribution. The analysis reveals a consistent ranking: normalization layers are most sensitive, followed by attention, and then MLP layers.

To connect the diagnostic to practice, the paper fine‑tunes one component at a time (freezing all others) on CIFAR‑10/100, CIFAR‑10‑C, Oxford‑Pets, and Flowers‑102. Sensitivity does not uniformly predict fine‑tuning gains across all components, but FFN‑Norm is a notable and practical exception: tuning only FFN‑Norm often matches or surpasses full fine‑tuning while requiring orders of magnitude fewer trainable parameters and less training memory. This yields a compelling, ultra‑cheap baseline for adaptation.

**Strengths:**

The paper proposes a simple, interpretable lens for analyzing submodule behavior under shift and executes a broad empirical study on ViTs that yields a consistent sensitivity ordering. The decision to couple analysis with an adaptation experiment makes the study more actionable. The FFN‑Norm result is a concrete takeaway for practitioners who need strong performance with very small trainable parameter budgets.

**Weaknesses:**

The claimed scope in the title suggests applicability to transformers in general, including NLP models, but the paper presents no empirical extension beyond vision. All experiments and the exact $W_2$ computation hinge on ViT‑style sequence lengths; long text sequences would require approximate OT and new validation. As written, the title over‑claims the generality of the contribution. A more accurate framing would restrict the title and claims to Vision Transformers or include at least a small NLP validation to justify the broader phrasing.


Beyond scope, the metric‑to‑practice connection is tenuous: $S(f)$ does not reliably indicate which component will help most when tuned, limiting the prescriptive power of the analysis outside the FFN‑Norm case. The OOD setting is primarily corruption‑based, omitting semantic and cross‑domain shifts where conclusions might change. Statistically, using a mean of ratios can be unstable when $W_2(\mu,\nu)$ is small; the paper does not report robustness checks (for example, trimmed means or ratio‑of‑means). Finally, the fine‑tuning configuration is narrow; modest optimizer and learning‑rate sweeps per component would better isolate component effects from recipe choices.

**Questions:**

1. How are $(\mu,\nu)$ pairs formed in $\mathcal{E}(D, D_{\text{ood}})$ (class‑matched, nearest‑neighbor, or random)? An ablation would help separate domain and class effects in $S(f)$.
2. How sensitive are the conclusions to alternative summaries (ratio‑of‑means vs. mean‑of‑ratios) or to trimming/winsorizing when denominators are small?
3. Is $f$ evaluated pre‑residual or post‑residual? Reporting both would clarify attribution in the presence of residual mixing.
4. Do the component‑wise fine‑tuning results persist under AdamW and modest learning‑rate sweeps? Even a small grid would reduce concerns about optimizer dependence.
5. Have you tried non‑corruption OOD (ImageNet‑A/R or cross‑domain datasets) for either sensitivity or fine‑tuning, even at small scale?
6. For prospective NLP extensions, which approximate optimal transport methods (e.g., Sinkhorn, low‑rank) are most promising for preserving the observed ViT component ordering, and why?

---

### Author Response · Authors · 2025-11-20
**Thanks for your time and insightful suggestions.**

Dear reviewers,

We would like to thank the reviewers for taking the time to carefully read our submission and providing helpful feedback. We appreciate the positive comments on the originality and practical use of our approach.

Overall, the reviewers are concerned about the sensitivity experiments outside of corrupted datasets and the lack of closer correlation between sensitivity and finetuning performance.

We observed during additional experiments that the sensitivity patterns remained valid across datasets (Cifar10/100, DomainNet, Cifar10-C). Regarding the correlation between sensitivity and performance, we note that we agree with the reviewers that the ideal case is to have metrics that can perfectly correlate but we believe this is hard to do since many other phenomena occur during the optimization process.

That being said, during our experiments for the rebuttal, we found other interesting phenomena that are connected to our current submission but that deserve a thorough treatment that will go beyond the rebuttal period. For this reason, we prefer to withdraw the paper to save reviewers and AC time and take the time to integrate those new findings for a more comprehensive study of sensitivity to distribution shifts and transformer adaptation to OOD data.

We again thank the reviewers for all their comments that were useful in improving our work and that gave us many interesting directions for future work.

Best regards,

The authors.

---

### Meta-Review · Area_Chair_kyYF · 2026-01-05

**Summary:**

Authors withdrew the paper during discussion period.

" For this reason, we prefer to withdraw the paper to save reviewers and AC time and take the time to integrate those new findings for a more comprehensive study of sensitivity to distribution shifts and transformer adaptation to OOD data."

**Reviewer Concerns:**

paper withdrawn

**Reviewer Scores:**

N/A

---

### Decision · Program_Chairs · 2026-01-26

Reject